# Screening for Antibiotics and Their Degradation Products in Surface and Wastewaters of the POCTEFA Territory by Solid-Phase Extraction-UPLC-Electrospray MS/MS

Sebastiano Gozzo [1], Samuel Moles [2,*], Katarzyna Kińska [1], Maria P. Ormad [2], Rosa Mosteo [2], Jairo Gómez [3], Francisco Laborda [4] and Joanna Szpunar [1]

1 Centre National de la Recherche Scientifique (CNRS), Institute of Analytical and Physical Chemistry for the Environment and Materials (IPREM), UMR 5254, Universite de Pau et des Pays de l'Adour, E2S, Hélioparc, 2, av. Pr. Angot, 64053 Pau, France
2 Water and Environmental Health Research Group, Institute of Environmental Sciences (IUCA), University of Zaragoza, Pedro Cerbuna 12, 50009 Zaragoza, Spain
3 Navarra de Infraestructuras Locales SA, av. Barañain 22, 31008 Pamplona, Spain
4 Group of Analytical Spectroscopy and Sensors (GEAS), Institute of Environmental Sciences (IUCA), University of Zaragoza, Pedro Cerbuna 12, 50009 Zaragoza, Spain
* Correspondence: sma@unizar.es

**Abstract:** A method based on UPLC-MS/MS (ultraperformance liquid chromatography—tandem mass spectrometry) was optimized for the analysis of a broad set of antibiotics and their metabolites in surface and wastewaters after their preconcentration by solid-phase extraction (SPE). The method was applied to the monitoring of the river basin of the POCTEFA (Interregional Programme Spain-France-Andorra) territory (Spain and France) in frame of a sampling campaign (2020–2021) including 40 sampling points, 28 of them corresponding to surface waters and 12 to wastewaters. In total, 21 antibiotics belonging to different families, i.e., ciprofloxacin, sulfamethoxazole, trimethoprim, azithromycin, and their metabolites were detected. A higher overall antibiotic contamination was observed in the Spanish part of the POCTEFA territory. Several metabolites of the target antibiotics, some of them supposed to be more toxic than their parent compounds, were identified in the entire sampling network. Fluoroquinolones and sulfamethoxazole, as well as their metabolites, presented the highest detection frequency both in wastewaters and surface waters, and, consequently, should be considered as target compounds in the monitoring of the water resources of the POCTEFA territory.

**Keywords:** antibiotics; solid phase extraction; LC-MS; metabolites; surface water; wastewater; POCTEFA

## 1. Introduction

Antibiotics are a group of drugs able to kill bacteria or inhibit their growth and division. They have become a target group of emerging pollutants due to their potential risks to public health and to the environment [1,2]. Already, in the late 1990s and 2000, the WHO convened a series of consultative groups, expert workshops, and consensus meetings to assess the growing public health threat of antimicrobial resistance resulting from the overuse of antibiotics and their release into the environment [2]. One of the main causes of antibiotic pollution is intensive farming and the excretion through faeces and urine during the free grazing of animals [3–5], followed by manure spreading on land [6,7], and contamination via runoff [3,8]. Hospitals are considered another source of emission of antibiotics, since high concentrations of them are usually found in their effluents [9,10]. Once released into the environment, antibiotics undergo different processes, such as dilution or concentration due to seasoning [10], dilution in surface water after waste water treatment plants (WWTPs) discharging [11], sorption to suspended particles [12], and degradation. The monitoring of the water environmental contamination levels by



antibiotics is of high importance to improve knowledge on their source pathways, transport, fate, and toxicity.

The studied territory belongs to the POCTEFA region. The POCTEFA territory covers an area of 115 583 km$^2$ and is populated by more than 15 million inhabitants [13]. However, we have studied the territory characterized by significant agricultural, intensive farming and industrial pressure. More precisely, the concerning territory is shown in Figure 1, and includes the provinces of Navarra, Huesca, Zaragoza, and Lleida in Spain and the departments of Pyrénées Atlantiques, Hautes Pyrénées, Pyrénées Orientales, Haute Garonne and Ariege in France.

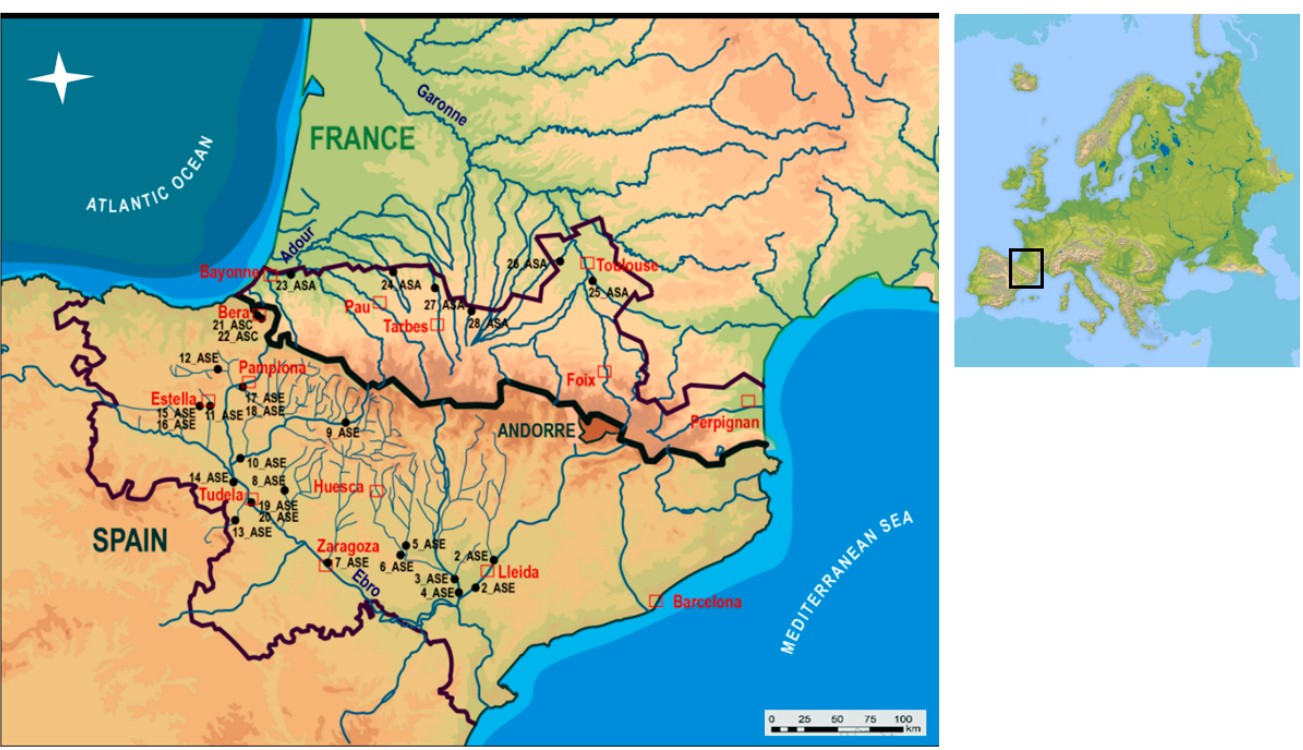

**Figure 1.** POCTEFA territory and surface sampling points.

A clear difference can be observed between the Spanish and the French parts of the POCTEFA territory regarding the consumption of veterinary antibiotics [14–17]. Between 2010 and 2018, antibiotics' sales patterns in Spain varied with the most popular being penicillins and tetracyclines followed by aminoglycosides, lincosamides, macrolides and sulfonamides. The information is difficult to interpret due to different strategies of data collection; nevertheless, a significant decline was observed in 2014, likely due to the adoption of the first "Spanish National Plan against Antibiotic Resistance" [18]. More complete data exist for France, where the sales of veterinary antibiotics have been monitored since 1999 by the French Agency for Food, Environmental and Occupational Health and Safety (ANSES) and its predecessors [18]. In 2015, France was the second largest consumer of veterinary pharmaceuticals worldwide, and the largest in Europe [15]. The level of exposure of animals to antibiotics, all routes and species combined, has decreased by 41.3% (from 1999 to 2019). In 2019, the overall exposure fell by 10.9% compared to the previous year and by 45.3% compared to 2011. Between 2018 and 2019 the change in exposure in France varied according to the species: −9.9% for cattle, −16.4% for pigs, −12.8% for poultry, and +1.5% for rabbits. Animals have been treated primarily with tetracyclines, penicillins, aminoglycosides, macrolides and polymyxins, followed by sulfonamides. The large decrease in antimicrobials used in animals in France is the result of collective action by all stakeholders to implement the French Action Plan 'EcoAntibio' 2012–2017. Another important source of antibiotics is human primary and hospital care. This use of antibiotics

in Spain is among the highest in Europe, while the consumption in France is 30% higher than the mean European rate [19]. Aware of a cross-border problem, European Union members agreed to the need to monitor 25 compounds, such as pharmaceuticals and pesticides, which are included in the Watch List of substances, recently updated in July of 2022 [20]. This list, in line with the European Action Plan "One Health", includes antibiotics such as sulfamethoxazole and trimethoprim, in order to deal with antibiotic resistance; however, this Watch List of substances does not include antibiotic metabolites. Consequently, studies of degradation products are necessary to assess the risk of antibiotics and improve regulations.

The impact of antibiotics on the environment is not a simple function of the consumed global amount. There is a difference in potency and doses between different drugs; new generation antibiotics are generally more efficient and require the administration of smaller doses of the active ingredient. An emerging issue is the consideration of the transformation products of the originally administered antibiotics, via biotic or abiotic processes [21,22]. As antibiotics belong to different groups of compounds and have different structures, elemental compositions, and physicochemical properties, there are no general rules governing their transformations [23]. Degradation products can show higher stability and toxicity than their parent compounds and possibly contribute to the development of antibiotic resistance genes [24,25].

Occurrence and risk assessment of antibiotics in surface waters have been the object of many recent studies [26–29]. LC-MS/MS using a triple quadrupole analyser has been the most currently applied method [12,21,22,27]. The studies have mostly concerned a single class of antibiotics, for instance fluoroquinolones [28] or sulfonamide antibiotics [10] with similar physicochemical properties. Five antibiotics (amoxicillin, clarithromycin, erythromycin, ofloxacin, sulfamethoxazole) could be determined among 40 emerging contaminants [27], and 46 antimicrobial drug residues in pond water [29]. The analyses usually targeted the marketed compounds without addressing their metabolites because of a lack of standards [10]. Non-targeted analyses of antibiotics or their degradation products in water by high resolution accurate mass spectrometry (HRAM) using TOF or Orbitrap analysers have been scarce [29].

The objectives of this work were: (i) revisiting the existing methodology for the simultaneous analysis of a large spectrum of antibiotics, together with their metabolites, at the detection limits, allowing the screening of surface waters and wastewaters; (ii) to apply this to obtain the first exploratory data on the contamination of the surface water and wastewaters of the POCTEFA territory; and (iii) to identify the principal sources of this contamination in areas with the highest density of livestock farms, hospitals and urban activities.

## 2. Experimental

### 2.1. Sampling

The sampling network included 40 sampling points, 28 of them corresponding to surface waters and 12 to wastewaters including hospital, urban and slaughterhouses effluents. As it can be seen in Figure 1, 6 sampling locations were situated in France and 22 in Spain. However, the exact locations of those concerning wastewaters are not disclosed for reasons of confidentiality.

Sampling was carried out according to the EPA 1694 Method [30], which is recommended for the analysis of pharmaceuticals and personal hygiene products. Water samples were collected in 1 L amber glass bottles, filled to overflowing to avoid the presence of air, and closed with a polypropylene cap and a polytetrafluoroethylene gasket. The samples were kept at 4 °C in the dark and filtered twice: first using 1 μm fiberglass filters provided by GVC, and then 0.45 μm nylon filters provided by GVC. Two sampling campaigns have been carried out, one in autumn 2020 and one in spring 2021.

## 2.2. Standards and Chemicals

Antibiotics standards were purchased from Sigma-Aldrich (Saint-Quentin-Fallavier, France), except for amoxicillin diketopiperazine and penicilloic acid (LGC, Molsheim, France). All the compounds were of high purity grade ($\geq$98%), except for florfenicol ($\geq$90%). The solvents (HPLC grade methanol and acetonitrile), formic acid ($\geq$95%), ammonium formate ($\geq$99%), ammonium acetate and ammonium bicarbonate ($\geq$99%) were purchased from Sigma-Aldrich.

Stock standard solutions were prepared by dissolving a weighed amount of the antibiotics in methanol, except for florfenicol, which was dissolved in ethanol, and fluoroquinolones, which were dissolved in 0.2% (*v/v*) hydrochloric acid in 50% (*v/v*) methanol. Working solutions were prepared by dilution with 50% methanol. Special precautions were taken for oxytetracycline, which was stored in the dark to avoid photodegradation [31]. Working solutions were prepared each month, while stock solutions were renewed every three months.

## 2.3. Solid-Phase Extraction (SPE)

A solid-phase extraction (SPE) vacuum system (CPI, Amsterdam, the Netherlands) was used to extract, clean up and preconcentrate the antibiotics from water samples. A further 10-fold concentration was achieved by solvent evaporation using a Concentrator FSC400D, dri-block from TECHNE (Fisher Scientific, Illkirch, France).

The experimental conditions were adapted from previous reports [13,21,22]. In brief, a 250 mL water sample was loaded at 5 mL min$^{-1}$ onto an Oasis HLB cartridge (diameter 47 mm, Waters, Guyancourt, France), preconditioned with 32 mL MeOH, and rinsed with 12 mL water and then with 12 mL water at pH 2.0 $\pm$ 0.5. The pH was determined by means of a Mettler Toledo InLab Expert Pro-ISM (Viroflay, France). The cartridge was dried for 5 min and eluted with 25 mL MeOH at 5 mL min$^{-1}$. A 1 mL aliquot of the resulting eluate was brought to dryness at 60 °C and reconstituted with 100 µL of 20% MeOH.

## 2.4. Targeted Analysis by LC-MS/MS

An Ultimate 3000 RSLC chromatographic system (ThermoFisher, Dreieich, Germany) was used for the separation of the antibiotics. A C18 (Accucore 100 $\times$ 2.1 mm, 2.5 µm) column was used. The antibiotics were eluted in gradient mode. Eluent A was 0.004 mM ammonium acetate/0.004 M ammonium formate in water. Eluent B was 0.004 mM ammonium acetate/0.004 M ammonium formate in a mixture containing 30% MeOH, 30% ACN, and 40% water. The pH of eluent A was adjusted with 0.3% of formic acid, while that of eluent B with 0.015 M of ammonium bicarbonate. These mobile phases allowed the sprayer voltage to be kept at less than 50 kV over the chromatographic run. The elution gradient was from 3% to 100% B in 16.5 min, 100% B for 8.5 min and back to 3% B within 3 min. Injection volume was 20 µL and flow rate 0.3 mL min$^{-1}$. Temperature was set at 35 °C.

The detection was performed by a Q Exactive Plus (ThermoFisher) high resolution mass spectrometer fitted with an IonMax ionization source and an HESI II probe. It was operated at sheath gas flow rate 50, auxiliary gas flow rate 20, sweep gas flow rate 1, capillary temperature (°C) 380, S-lens RF level 50 and aux gas heater temperature (400 °C).

MS data acquisition was performed in positive mode using parallel reaction monitoring (PRM). The m/z isolation window was 0.5 Da, resolution 17,500, and AGC target $1 \times 10^5$. The precursor ions and two product ions per compound were monitored. These are listed, together with the collision energies used, in Table 1.

The criteria for the confirmation of species identity were (i) a precursor mass within 5 ppm mass tolerance, (ii) at least 2 to 3 isotopes of the isotopic pattern within 5 ppm, (iii) minimum intensity threshold $\geq$3, and (iv) chromatographic RT not to exceed 0.5 min. These criteria fulfil the FDA Acceptance Criteria for Confirmation of Identity of Chemical Residues using Exact Mass Data within the Office of Foods and Veterinary Medicine (September 2015) [32].

**Table 1.** MS detection parameters used in the quantitative analysis of the antibiotic species.

| Group | Antibiotic | CAS | RT (min) | Parent Ion (m/z) [M + H]$^+$ | CE (eV) | Product Ion 1 | Product Ion 2 | LOD (ng L$^{-1}$) | LOQ (ng L$^{-1}$) |
|---|---|---|---|---|---|---|---|---|---|
| ß-Lactamase | amoxicillin | 26,787-78-0 | 8.99 | 366.1118 | 10 | 349.0853 | 208.0427 | 0.152 | 0.500 |
| | ampicillin | 69-53-4 | 5.93 | 350.1169 | 20 | 192.0478 | 106.051 | 0.021 | 0.071 |
| | diketopiperazine | 94,659-47-9 | 9.36 | 366.1118 | 10 | 207.0764 | 160.0427 | 0.300 | 0.100 |
| | penicilloic acid | 210,289-72-8 | 5.50 | 384.1224 | 10 | 367.0959 | 323.1058 | 0.015 | 0.05 |
| diaminopyrimidine | trimethoprim | 738-70-5 | 9.06 | 291.1452 | 50 | 261.0982 | 230.1162 | 0.026 | 0.085 |
| fluoroquinolone | ciprofloxacin | 85,721-33-1 | 9.91 | 332.1405 | 65 | 249.0670 | 231.0564 | 0.028 | 0.094 |
| | enrofloxacin | 93,106-60-6 | 10.47 | 360.1718 | 35 | 316.1820 | 245.1085 | 0.041 | 0.135 |
| | moxifloxacin | 354,812-41-2 | 12.64 | 402.1824 | 40 | 384.1718 | 341.1534 | 0.011 | 0.035 |
| | norfloxacin | 70,458-96-7 | 9.66 | 320.1405 | 35 | 276.1507 | 233.1085 | 0.015 | 0.050 |
| lincosamide | lincomycin | 154-21-2 | 8.44 | 407.2210 | 25 | 359.2177 | 126.1277 | 0.025 | 0.082 |
| macrolide | azithromycin | 83,905-01-5 | 13.99 | 749.5158 | 25 | 591.4215 | 158.1176 | 0.020 | 0.067 |
| | clarithromycin | 81,103-11-9 | 20.91 | 748.4842 | 20 | 590.3899 | 158.1176 | 0.018 | 0.059 |
| | clarithromycin N-oxide | 118,074-07-0 | 20.87 | 764.4791 | 28 | 606.3848 | 123.0804 | 0.012 | 0.038 |
| sulfonamide | dapsone | 80-08-0 | 9.94 | 249.0692 | 35 | 156.0114 | 108.0444 | 0.029 | 0.095 |
| | sulfacetamide | 144-80-9 | 5.69 | 215.0485 | 35 | 156.0114 | 108.0444 | 0.049 | 0.163 |
| | sulfadiazine | 68-35-9 | 6.71 | 251.0597 | 30 | 156.0114 | 108.0444 | 0.015 | 0.050 |
| | sulfadoxine | 2447-57-6 | 11.31 | 311.0809 | 40 | 156.0114 | 108.0444 | 0.012 | 0.038 |
| | sulfamerazine | 127-79-7 | 8.14 | 265.0754 | 35 | 190.0274 | 156.0114 | 0.023 | 0.075 |
| | sulfapyridine | 144-83-2 | 6.82 | 250.0645 | 35 | 184.0869 | 156.0114 | 0.015 | 0.050 |
| | sulfamethoxazole | 723-46-6 | 11.19 | 254.0594 | 40 | 156.0114 | 108.0444 | 0.008 | 0.025 |
| | sulfamethoxypyridazine | 80-35-3 | 10.73 | 281.0703 | 30 | 156.0114 | 126.0662 | 0.058 | 0.197 |
| tetracycline | oxytetracycline | 79-57-2 | 9.75 | 461.1555 | 20 | 426.1183 | 381.0605 | 0.028 | 0.094 |

Abbreviations: RT = retention time, CE = collision energy, LOD = Limit of Detection, LOQ = Limit of Quantification. Quantification was carried out using XCalibur 4.2 software on the basis of an 8-point matrix-matched calibration curve (R $\geq$ 0.998).

*2.5. Detection and Identification of Antibiotic Degradation Products by LC-High Resolution MS/MS*

The UPLC column mentioned above was used. Eluent A was 0.1% formic acid and eluent B was 0.1 formic acid in MeOH:ACN (1:1, *v/v*). The elution gradient was from 10% to 95% B within 9.5 min, 95% B for 2 min, and back to 10% B within 1 min. The column was reconditioned with 10% B for 3 min. Injection volume was 20 µL, flow rate of 0.3 mL/min, and temperature 35 °C. Sheath gas flow rate was 50, auxiliary gas flow rate 20, sweep gas flow rate 1, capillary temperature 380 °C, S-lens RF level 50 and auxiliary gas heater temperature 400 °C.

MS and $MS^2$ data acquisition was performed in positive move in full MS-dd$MS^2$ (top5, with an exclusion list extracted from the last blank before analysing the samples). The settings were full MS-SIM HR 70,000, AGC target 1e6, scan range 100–800 m/z, dd-$MS^2$ resolution 35,000, AGC target $1 \times 10^5$, loop count 5, m/z isolation window 0.5, (N)CE 15 and 45. All fragmentation information could be obtained using one full scan and without sample re-injection in $MS^2$ mode.

The compounds were identified using Compound Discoverer software. For the investigation of the transformation products, a list was assembled from the literature [23,25,33]. For the confirmation by CD, an already existing ThermoFischer Scientific Workflow [33–38] was applied with the addition of Fragment Ion Search (FISh) processing being used [36].

For the compounds for which no analytical standards were available, the identification criteria were (i) precursor mass within 5 ppm mass tolerance, (ii) at least 2 to 3 isotopes of the isotopic pattern within 5 ppm, and (iii) minimum intensity $\geq 3$. The criteria for MS2 spectra were 5 ppm mass tolerance and an FISh score above 25%. According to the guidelines formulated in the literature on identifying small molecules via high resolution mass spectrometry [37], this corresponds to the confidence identification level 2.

*2.6. Method Validation*

Individual recoveries for each compound at three different concentration levels with the corresponding relative standard deviations matrix effect, linearity, LOQs and LODs have been included in the Appendix A information (Tables A1 and A2).

*2.7. ANOVA Test*

ANOVA test was implemented to determine the existence of significant differences between antibiotic concentrations in Spanish and French rivers, a *p*-value of 0.05 was selected (*p*-value < 0.05 for significant differences). The data was treated with Microsoft Excel.

## 3. Results

### 3.1. Quantitative Analysis of Antibiotics: Method Development and Validation

Reversed-phase UPLC using double-gradient (pH and organic solvent) has been an established approach to the separation of antibiotics [39–41]. The use of electrospray ionization imposes the use of volatile buffers and reduces concentration of the salts, which might increase sprayer voltage leading to rim emission or corona discharge.

The LC coupling to high-resolution hybrid quadrupole-Orbitrap mass spectrometry allowed the identification and quantification of compounds in one chromatographic run. A parallel reaction monitoring (PRM) scan mode strategy was used; it consists of the isolation of a targeted precursor in Q1, and then all generated MS/MS fragment ions are recorded in parallel with the characteristics of a full scan, accurate mass, and high-resolution. A baseline separation of the 21 antibiotics and 3 degradation products (amoxicillin diketopiperazine and penicilloic acid as well as clarithromycin N-oxide), for which commercial standards could be purchased, has been achieved, as shown in Figure 2.

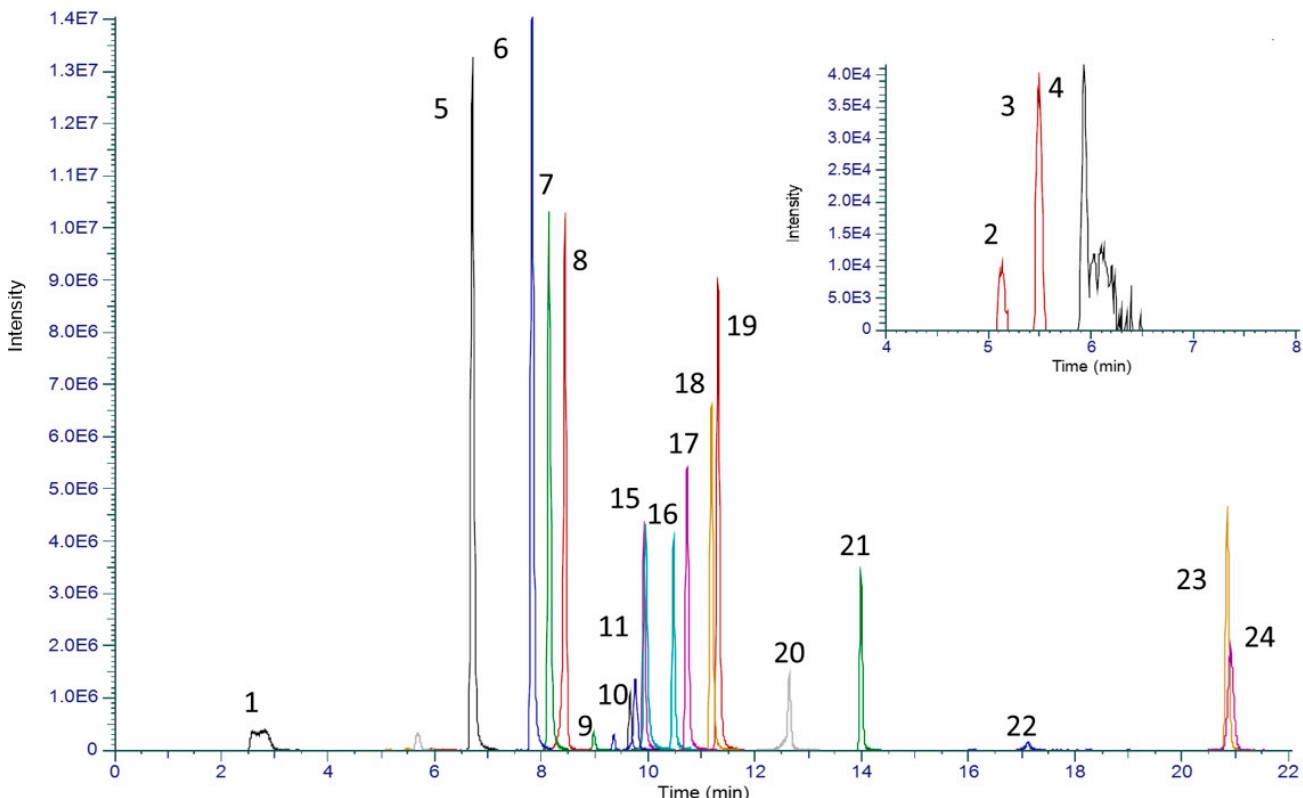

**Figure 2.** LC-MS chromatogram obtained for the standards (5 µg L$^{-1}$): 1—florfenicol amine, 2—penicilloic acid, 3—sulfacetamide, 4—amoxicillin, 5—sulfadiazine, 6—sulfapyridine, 7—sulfamerazine, 8—lincomycin, 9—ampicillin, 10—trimethoprim, 11—amoxicillin diketopiperazine, 12—norfloxacin, 13—oxytetracycline, 14—ciprofloxacin, 15—dapsone, 16—enrofloxacin, 17—sulfamethoxypyridazine, 18—sulfamethoxazole, 19—sulfadoxine, 20—moxifloxacin, 21—azithromycin, 22—erythromcin, 23—clarithromycin N-oxide, 24—clarithromycin.

The relatively low levels of the antibiotics found in the freshwater samples required an SPE protocol to extract and preconcentrate them; the procedure served also to clean-up (more polluted) wastewater samples. Indeed, a problem encountered in the development of targeted analytical methods for species over a wide range of properties is that a balance must be kept between the ability to preconcentrate them without simultaneously extracting too many other compounds that result in a heavy matrix. The SPE extraction and preconcentration was optimised by assessing the effect of several variables including sample pH, the type of cartridge, cleaning step, and elution solvents, as well as the evaluation of different preconcentration factors on possible interferences. Finally, the method used was based on the EPA1694 with some modification, in particular, no EDTA was used.

A matrix-matched calibration was used to perform the quantification; a calibration curve was constructed with eight points using a least-square linear regression (R ≥ 0.998). Recovery was measured by spiking three different concentrations (0.01, 0.05 and 0.1 ng L$^{-1}$) in each kind of water matrix and the analysis of the resulted solutions after the entire analytical procedure mentioned before. The results are presented in Table A1. The instrumental repeatability of the LC–QqLIT–MS equipment was calculated through six consecutive injections of a standard antibiotic mixture solution corresponding to a concentration of 500 ng L$^{-1}$. Three blank samples for each matrix were also evaluated to avoid overestimations in the calculation of the recovery. Instrumental detection limits (ILODs) were experimentally calculated from the injection of the standard solution with a concentration corresponding to the lowest used to build the calibration curves (in this study, 0.05 ng L$^{-1}$). Method LODs and LOQs (Table A2) were experimentally calculated from the analysis of the spiked water samples based on a signal to noise ratio of 3 and 10, respectively.

Recoveries higher than 50% were obtained for all the antibiotics, except for clarithromycin and clarithromycin N-oxide (40 < R% < 50). The highest, fully quantitative, recoveries were obtained for trimethoprim and azithromycin. In general, β-lactam antibiotics showed lower recoveries than fluoroquinolones, which exhibited lower recovery than sulfonamides. Instrumental repeatability was calculated through six consecutive injections of standard antibiotic spiked in matrix water. LODs and LOQs (given in Table 1) were calculated using the standard deviation of the lowest point (analysed 20 times), divided by the slope, and multiplied respectively for 3 and 10.

Figure 3 shows an example chromatogram for hospital (Figure 3a,b) and bird slaughterhouse samples. It can be seen how the chromatograms are quite different. The hospital effluent presents a broad variety of antibiotics detected from different families (more than 15 peaks). However, the slaughterhouse effluent presented fewer than 10 antibiotics, which are individually in higher concentrations, although the total antibiotic concentration of the hospital effluent were about five times higher than the slaughterhouse effluent. Moreover, in the case of the slaughterhouse, they specifically correspond to veterinary antibiotics (e.g., enrofloxacin).

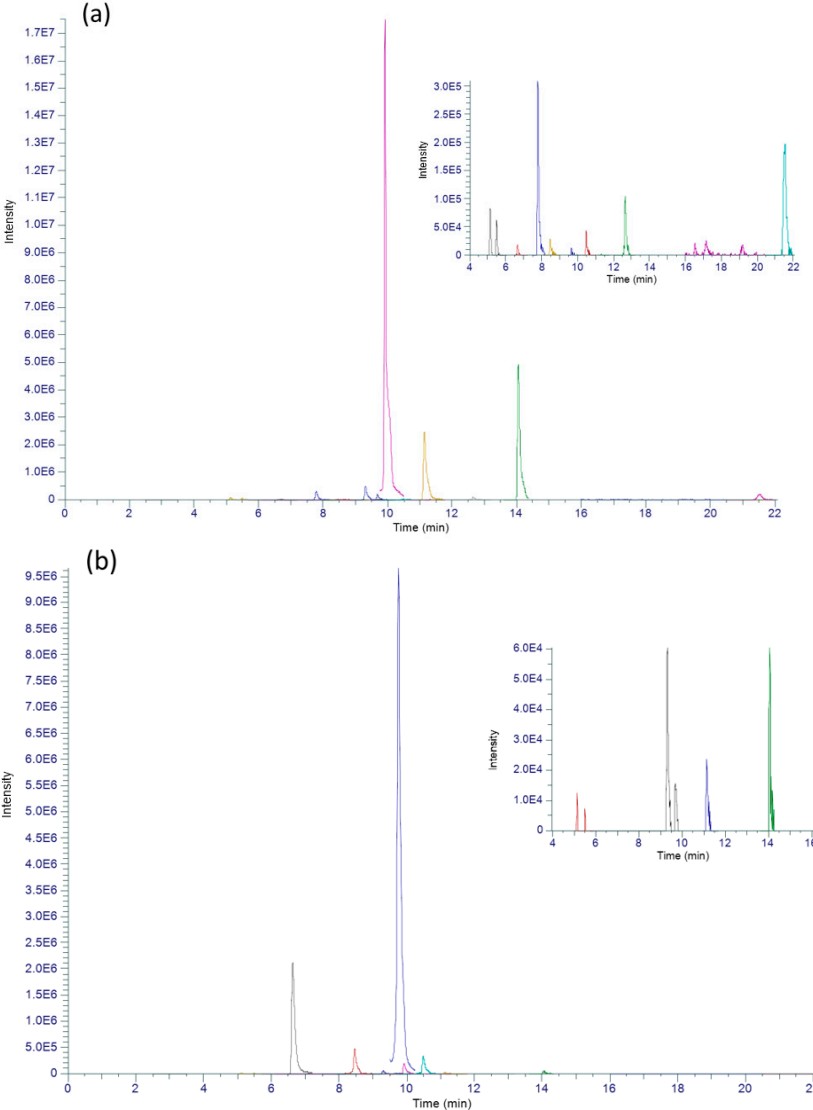

**Figure 3.** LC-MS example chromatograms obtained for (**a**) hospital and (**b**) bird slaughterhouse samples.

### 3.2. Quantification of Antibiotics in Surface Waters

The frequency of the detection and concentration range of target antibiotics obtained for surface waters are summarized in Table 2. The results are discussed in the context of the data reported for the quantification of antibiotics in European surface waters, and summarized in Table 3.

**Table 2.** Frequency of the detection and concentration range of target antibiotics in surface waters in comparison with the maximum concentration reported in Europe.

| Group | Antibiotic | Frequency of Detection (%) | Concentration Range (ng L$^{-1}$) | C$_{max}$ (ng L$^{-1}$) * |
|---|---|---|---|---|
| ß-lactamase | amoxicillin | 4 | LOQ-8.0 | 522 (UK) [40] |
| | ampicillin | 14 | 15.7–79.6 | 26 (Germany) [41] |
| | diketopiperazine | - | <LOQ | not given |
| | penicilloic acid | - | <LOQ | not given |
| fluoroquinolone | ciprofloxacin | 29 | LOQ-33.6 | 9660 (France) [42] |
| | enrofloxacin | 89 | 11.8–970.0 | 210 (Portugal) [43] |
| | moxifloxacin | 7 | 1.4–9.8 | 210 (Spain) [44] |
| | norfloxacin | 4 | LOQ-3.2 | 160 (France) [45] |
| lincosamide | lincomycin | 29 | 0.9–70.7 | 250 (Italy) [46] |
| macrolides | azithromycin | 21 | 1.7–67.0 | 1600(Croatia) [47] |
| | erythromycin | - | <LOQ | 1700 (Germany) [48] |
| | clarithromycin | 7 | 3.8–7.0 | 2330 (France) [42] |
| | clarithromycin n-oxide | 4 | LOQ-4.0 | not given |
| sulfonamide | dapsone | 4 | 1.1–30.7 | not given |
| | sulfadiazine | 61 | 0.4–48.3 | 2400 (Croatia) [47] |
| | sulfamerazine | 4 | LOQ-8.6 | 11,000 (Croatia) [47] |
| | sulfamethoxazole | 68 | LOQ-63.9 | 11,000 (Spain) [49] |
| | sulfadaxone | - | <LOQ | not given |
| | sulfapyridine | 11 | 1.4–48.8 | 12,000 (Spain) [50,51] |
| tetracycline | oxytetracycline | 4 | LOQ-7.8 | not given |
| diaminopyrimidine | trimethoprim | 64 | 2.5–106.0 | 11,000 (Croatia) [47] |

* Abbreviations: C$_{max}$ = Maximum concentration reported for target antibiotics in the surface waters of Europe, LOQ = Limit of Quantification.

Target antibiotics have been detected in all of the surface water samples with the exception of erythromycin, diketopiperazine and penicilloic acid. There is, however, no single antibiotic that would be detected in all of the samples, which contrasts with the omnipresence of some antibiotics observed elsewhere [52–54]. According to Figure 4, the detection frequency of antibiotics ranged from 4% to 89%. The highest was observed for enrofloxacin (89%), sulfamethoxazole (68%), trimethoprim (64%), sulfadiazine (61%) and ciprofloxacin (29%). Moreover, enrofloxacin showed not only the highest detection frequency but also the highest maximum concentration, as can be seen in Figure 4. This observation is similar to that made in Portuguese surface waters [55,56], where sulfamethoxazole was the most frequently (92%) detected, followed by ciprofloxacin (75%), while both compounds appeared in many sampling points at higher concentrations.

It is interesting to compare the concentrations found in this study with the maximum concentrations (Table 2) reported for antibiotics in surface waters in Europe (Table 3). The most striking is the case of enrofloxacin of which the maximum concentration measured in this study reaches 970 ng L$^{-1}$, which is almost five times higher than the maximum concentration of this antibiotic reported so far in Europe (212 ng L$^{-1}$ in Portugal) [43]. The exposure to this antibiotic in the POCTEFA territory is corroborated with its highest frequency of detection. A similar observation was made for ampicillin, for which the highest concentration found in this study (79.6 ng L$^{-1}$) is ca. three times higher than reported elsewhere in Europe (26 ng L$^{-1}$) [41].

**Table 3.** Concentrations of antibiotics found in wastewaters.

| Group | Antibiotic | Frequency of Detection (%) | Concentration Range (ng L$^{-1}$) |
|---|---|---|---|
| ß-lactamase | amoxicillin | 8 | 7.0–15.0 |
| | ampicillin | 25 | LOQ-26.0 |
| | diketopiperazine | - | <LOQ |
| | penicilloic acid | - | <LOQ |
| fluoroquinolone | ciprofloxacin | 75 | 3.8–172.7 |
| | enrofloxacin | 61 | 48.3 |
| | moxifloxacin | 42 | 2.6–46.9 |
| | norfloxacin | 25 | LOQ-3.2 |
| lincosamide | lincomycin | 50 | LOQ-26.0 |
| macrolides | azithromycin | 92 | 1.8–144.0 |
| | erythromycin | 8 | LOQ-43.0 |
| | clarithromycin | 50 | 6.6–62.7 |
| | clarithromycin N-oxide | - | <LOQ |
| sulfonamide | dapsone | 8 | LOQ-30.7 |
| | sulfadiazine | 42 | LOQ-103.0 |
| | sulfamerazine | - | <LOQ |
| | sulfamethoxazole | 92 | 5.9–256.0 |
| | sulfapyridine | 42 | 2.8–12.6 |
| tetracycline | oxytetracycline | 8 | 535–2670 |
| diaminopyrimidine | trimethoprim | 75 | 1.4–122.9 |

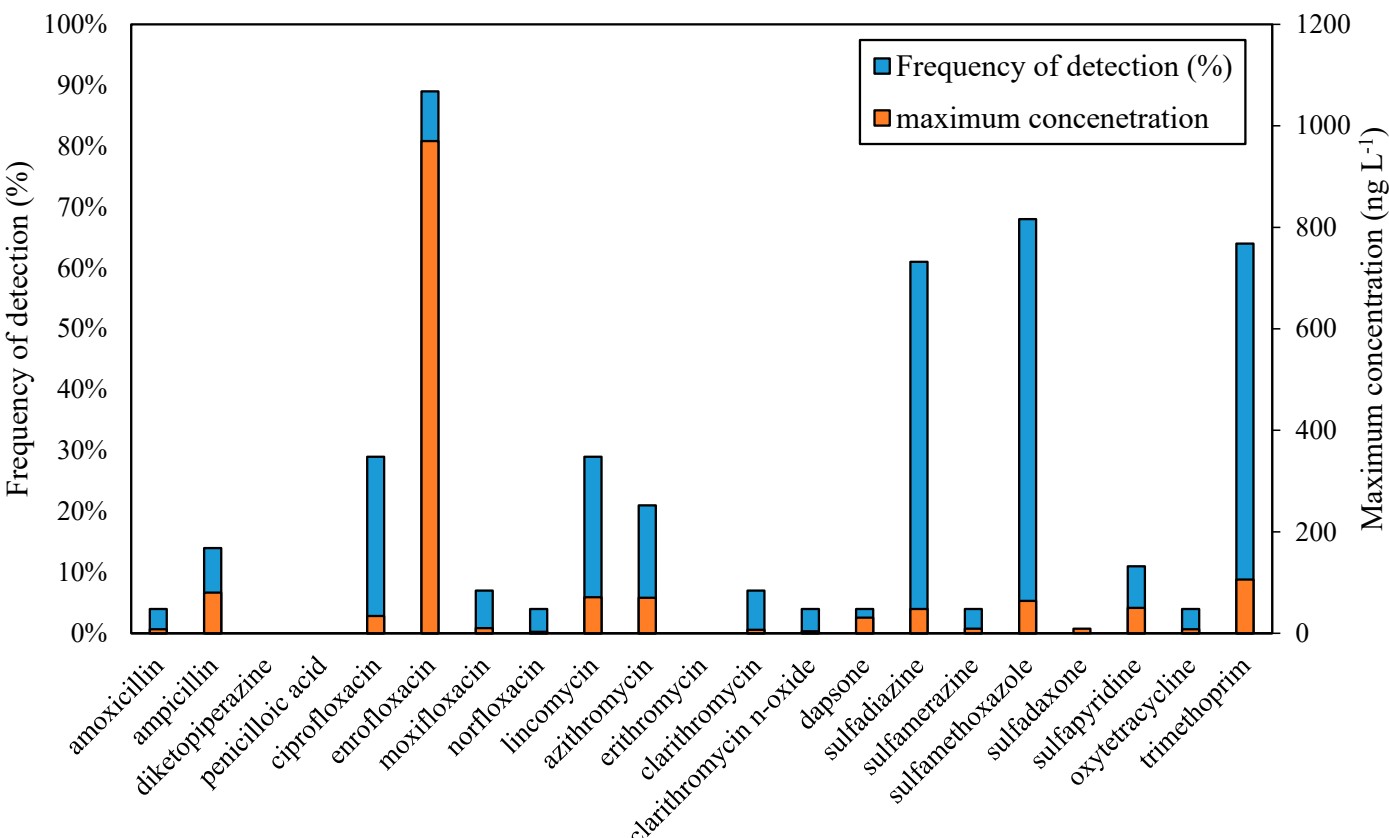

**Figure 4.** Detection frequency (blue bar) and maximum concentration (orange bar) of selected compounds in surface water.

On the other hand, the concentrations found for the other antibiotics (trimethoprim, sulfadiazine, sulfamethoxazole, azithromycin) are much lower than the concentrations reported elsewhere for these compounds. For example, for trimethoprim and sulfamethoxazole, which were the next most frequently detected antibiotics, the maximum concentrations are 106 ng L$^{-1}$ for trimethoprim and 63.9 ng L$^{-1}$ for sulfamethoxazole, while in Europe the maximum concentration detected for these antibiotics were two orders of magnitude higher: 11,000 ng L$^{-1}$ in Croatia [47] for trimethoprim and 11,000 ng L$^{-1}$ in Spain [47]. This observation is similar for sulfadiazine, azithromycin and ciprofloxacin, of which the concentrations measured ranged from 33.6 ng L$^{-1}$ to 67.0 ng L$^{-1}$, while their maximum concentrations reported in Europe were at least two orders of magnitude higher. Lincomycin showed a concentration of 70.7 ng L$^{-1}$, whereas the maximum concentration in European surface waters was 248.9 ng L$^{-1}$.

The ANOVA test was implemented to determine the existence of significant differences between Spanish and French rivers, a *p*-value of 0.05 was selected according to the literature [53]. Regarding the results, which are summarized in Table A3, significant differences ($p < 0.05$) have been found between the concentration of target antibiotics in French and Spanish rivers. The total concentration of the target antibiotics in French rivers reaches about 10 ng L$^{-1}$ in the Adour River, near Bayonne (23_ASA/G), while the total concentration of the studied antibiotics in a sampling point in Spain exceeds 1000 ng L$^{-1}$. This could be due to the fact that the use of antibiotics in Spain is the highest in the EU/EEA [54,55], or to the later adoption of the plan against antibiotic resistance in Spain than in France.

These observations are consistent with the data reported on the occurrence of antibiotics in Spain and France, which were summarized in Table A4. The concentrations of antibiotics measured in Spain are generally higher than in France with the exceptions of ciprofloxacin and clarithromycin. This conclusion should be taken with caution, however, as there have been many more studies concerning the presence of antibiotics in Spain (ca. 171) than in France (ca. 30) [56]. The maximum total concentration of the target antibiotics measured in a singular sampling point reaches 1247.6 ng L$^{-1}$, which is similar to the concentration reported in Chinese rivers for these antibiotics [52].

### 3.3. Concentrations of Antibiotics Measured near to Wastewater Treatment Plants (WWTP)

The results (concentration range and frequency of detection) obtained for the quantification of antibiotics in wastewaters are summarized in Table 3.

Target compounds have been detected in all of the selected water samples with the exceptions of clarithromycin N-oxide, sulfamerazine, diketopiperazine and penicilloic acid. They present a high frequency of detection from 8% to 92%. The highest frequency of detection has been observed for the antibiotics: azithromycin (92%), sulfamethoxazole (92%), trimethoprim (75%), ciprofloxacin (75%), enrofloxacin (61%) and clarithromycin (50%). Comparing these results with those obtained in the previous section, it can be seen that ciprofloxacin, enrofloxacin, sulfamethoxazole and trimethoprim are the antibiotics that appear in highest chronicity in both, surface waters and wastewater effluents. Among the antibiotics investigated, ciprofloxacin was predominant in WWPT influent and effluents. Sulfamethoxazole was also very abundant in the influents (>83%) and effluents (>79%).

In the present research, oxytetracycline (2670 ng L$^{-1}$), azithromycin (144 ng L$^{-1}$), ciprofloxacin (176 ng L$^{-1}$), and sulfamethoxazole (256 ng L$^{-1}$) appeared in the highest concentrations. Other authors also detected the presence of ciprofloxacin as the antibiotic predominant in Greece (48), presenting concentrations up to 591 ng L$^{-1}$. This work also suggests that sulfamethoxazole was very abundant in wastewater (>83%) and effluents (>9%); however, this sulfonamide antibiotic was found in relatively low concentrations (<137.9 ng L$^{-1}$) in influents and (<43 ng L$^{-1}$) effluents [57]. Other studies have evaluated the occurrence of pharmaceuticals in the effluent of wastewater treatment plants in Italy, reporting again that although sulfamethoxazole and ciprofloxacin presented a frequency of detection of 100%, these compounds appeared in relatively low concentrations in ef-

fluents [57]; more precisely, they reported concentrations of ciprofloxacin ranging from 10–500 ng $L^{-1}$ [58] and from 35–185 ng $L^{-1}$ of sulfamethoxazole.

Regarding the load of antibiotics, in this research, azithromycin, trimethoprim and oxytetracycline showed the maximum concentrations, ranging from 144 ng $L^{-1}$ for azithromycin to 2670 ng $L^{-1}$ for oxytetracycline. Although this antibiotic was only detected in 8% of the wastewater samples, this concentration can be due to a punctual emission.

Table 4 shows a summary of a European study, which reported antibiotic concentrations in the WWTPs effluent of seven European countries (Portugal, Spain, Ireland, Cyprus, Germany, Finland, and Norway). The compounds with the highest loads in the countries studied in this report were macrolides and fluoroquinolones [59–63]. As a result, ciprofloxacin was selected as a marker of antibiotic pollution and was suggested to be used for widespread temporal and geographical characterization of environmental water or WWTP effluents by other authors [60].

**Table 4.** Summary of the results in a recent study of 7 WWTPs in the European scenario [60].

| Group | Antibiotic | $C_{max}$ (ng $L^{-1}$) | Country |
|---|---|---|---|
| fluoroquinolones | ciprofloxacin | 1436 | Portugal |
| | enrofloxacin | 176 | Spain |
| macrolides | azithromycin | 1577 | Portugal |
| | clarithromycin | 337 | Ireland |
| sulfonamides | sulfamethoxazole | 177 | Spain |
| diaminopyrimidine | trimethoprim | 330 | Finland |

Abbreviations: $C_{max}$ = maximum concentration.

### 3.4. Analysis for Antibiotic Degradation Products

β-lactams are structurally characterized by the β-lactam nucleus, which is susceptible to cleavage, e.g., by high temperatures, light, extremes in pH, metal ions, oxidizing and reducing agents [61], as well as enzymatic and biological degradation. Consequently, low environmental exposure levels of β-lactams are expected, despite their high consumption [13]. For instance, in our work, the degradation of amoxicillin was found to be matrix-dependent with the loss of 19%, 37% and 75% of the original compound during one-week storage at room temperature for tap, fresh and wastewater, respectively. Interestingly, one of the major degradation products of amoxicillin, amoxicillin diketopiperazine, has the same m/z (366.1) as the parent species; however, it eluted at a different RT and, thus, both of them could be easily distinguished. Although the hydrolysis of the β-lactam ring results in a loss of the antibiotic activity, the identification of their transformation products and the study of their occurrence, fate, efficiency, and persistence in the environment are essential for proper risk assessments [64–67]. Two degradation products of amoxicillin (penicilloic acid and diketopiperazine), for which commercial standards are available, could be quantified.

The metabolites for which standards were unavailable were searched for, either in a targeted way based on the literature information or using exploratory workflows. The species found, together with their formulas and masses, are summarized in Table 5; they have been previously reported in environmental waters [23,33,68,69]. The identified degradation products had to show at least two fragments matching the ones of the parent molecule and an additionally two fragments with a delta mass matching the identified transformation. According to the guidelines formulated by other authors on identifying small molecules via high resolution mass spectrometry [37], it corresponds to the confidence identification level 2.

**Table 5.** The antibiotic degradation products detected.

| Group | Antibiotic | Degradation Product | RT (min) | Formula | Exact Mass | Δ ppm | Ref. |
|---|---|---|---|---|---|---|---|
| diaminopyrimidine | trimethoprim | 4-desmethyl-TMP | 4.16 | $C_{13}H_{17}N_4O_3$ | 277.1294 | −0.5 | [64] |
| fluoroquinolone | enrofloxacin | enrofloxacin 5Bwt | 5.80 | $C_{17}H_{21}FN_3O_3$ | 334.1558 | −1.37 | [66] |
| macrolides | azithromycin | azithromycin double cleavage | 5.90 | $C_{22}H_{44}NO_7$ | 434.3110 | −0.38 | [65] |
| sulphonamide | sulfadiazine | N-acetyl sulfadiazine | 4.98 | $C_{12}H_{13}N_4O_3S$ | 293.0701 | −0.73 | [23,33] |
| | sulfamethoxazole | N-acetyl sulfamethoxazole | 6.63 | $C_{12}H_{14}N_3O_4S$ | 296.0698 | −0.6 | [23] |

Abbreviations: RT = retention time.

The results of the frequency of detection of antibiotic degradation products in surface and wastewaters are summarized in Table 6.

**Table 6.** Frequency of detection of antibiotic degradation products in surface and wastewaters.

| Group | Antibiotic | Degradation Product | Frequency of Detection (%) | |
|---|---|---|---|---|
| | | | Surface Water | Wastewater |
| -lactamates | amoxicillin | penicilloic acid | - | - |
| | | diketopiperazine | - | - |
| macrolides | azithromycin | azithromycin double cleavage | 57 | 50 |
| diaminopyrimidines | trimethoprim | 4-desmethyl-TMP | 7 | 50 |
| fluoroquinolones | enrofloxacin | enrofloxacin 5Bwt | 14 | 58 |
| sulphonamides | sulfadiazine | N-acetyl sulfadiazine | 11 | 8 |
| | sulfamethoxazole | N-acetyl sulfamethoxazole | 11 | 67 |

Note that even though some antibiotics were hardly detected in surface waters, their metabolites were. For example, azithromycin showed a detection frequency of 21%, while its degradation product with double cleavage was detected almost three times more often (57%). This corroborates earlier indications [23,67] that azithromycin is transformed in the aquatic environment.

The detection frequency of the metabolites of sulfamethoxazole (N-acetyl-sulfamethoxazole) and enrofloxacin (enrofloxacin-5Bwt) in wastewaters follows that of their parent compounds. Note that enrofloxacin-5Bwt shows a frequent presence in wastewaters (58%) but is less frequently detected in surface river waters (14%). This can be attributed to the degradation of fluoroquinolones in the biological processes applied in wastewater treatment and/or to the degradation of enrofloxacin in the environment [70–74]. Amoxicillin metabolites, penicilloic acid and diketopiperazine, for which standards exist and which were monitored in a targeted way, were not detected in any sample.

As a result, the antibiotics included in the EU's Watch List have not only been observed in surface waters from Spain and France, but their metabolites have too, which highlights the high pressure to which the environment in this study area is subjected.

## 4. Conclusions

The developed SPE-LC-MS/MS method allowed the determination of 21 antibiotics and the detection of their metabolites in the POCTEFA territory. There were significant differences between the concentration of antibiotics in surface waters of France and Spain, reflecting the larger use of antibiotics in Spain. In surface waters, enrofloxacin, sulfamethoxazole and trimethoprim appeared at the highest concentrations and showed the highest frequencies of detection. In wastewaters, oxytetracycline, azithromycin, ciprofloxacin, and sulfamethoxazole appeared in the highest concentrations. Ciprofloxacin, enrofloxacin, sulfamethoxazole and trimethoprim were the antibiotics that showed the highest chronicity in both surface waters and wastewater effluents. Some degradation products of antibiotics, e.g., azithromycin, presented a higher frequency of detection than their parent compounds, suggesting degradation occurs in WWTPs and in the environment.

It is necessary to continue with the monitoring of antibiotics in the waters of the POCTEFA territory. Since this territory experiences high pressure from livestock antibiotics, reducing both intensive farming and the use of livestock antibiotics could be crucial to warrantee the quality of surface waters, especially in Spain. Moreover, the release of citizen awareness campaigns about the correct use of antibiotics and antibiotic disposal would also be very interesting. These actions should be in the framework of a legal regulation that limits the concentration of the most detected antibiotics (e.g., ciprofloxacin) in collector discharges and/or in WWTPs before discharging into the environment.

**Author Contributions:** Conceptualization, R.M.; Data curation, S.M. and K.K.; Formal analysis, S.G.; Investigation, J.G.; Methodology, M.P.O. and J.S.; Project administration, J.S.; Supervision, M.P.O., R.M. and J.S.; Writing—original draft, S.M.; Writing—review and editing, K.K., J.S. and F.L. All authors have read and agreed to the published version of the manuscript.

**Funding:** This research was funded by the project EFA 1983/16 OUTBIOTICS co-financed by the European Regional Development Fund (ERDF) through the European Program of territorial cooperation POCTEFA 2014-20 (INTERREG POCTEFA). K.K. acknowledges the Région Aquitaine post-doctoral grant (Contrat Nr 206406). S.G. is grateful for the technical help of Simon Godin and Ange Angaïts. S.M. appreciates the grant Margarita Salas funded by the European Union-NextGenerationEU.

**Conflicts of Interest:** The authors declare no conflict of interest.

## Appendix A

**Table A1.** Recovery and matrix effect data for the developed procedure; * WWTP—water from wastewater treatment plant.

| Group | Antibiotic | Recovery (%) | | | Matrix Effect (%) | | | Linearity |
|---|---|---|---|---|---|---|---|---|
| | | 0.01 | 0.05 | 0.1 | River Water | Outlet-WWTP * | Inlet-WWTP * | |
| florfenicol metabolite | florfenicol amine | 70 ± 18 | 75 ± 14 | 72 ± 17 | 90 | 101 | 106 | 0.998 |
| ß-Lactamase | amoxicillin | 51 ± 18 | 55 ± 12 | 58 ± 15 | 85 | 78 | 70 | 0.998 |
| | ampicillin | 51 ± 15 | 55 ± 12 | 55 ± 11 | 77 | 75 | 71 | 0.998 |
| ß-Lactamase AMX degradation product | diketopiperazine | 60 ± 11 | 62 ± 13 | 63 ± 11 | 76 | 78 | 70 | 0.998 |
| | penicilloic acid | 68 ± 18 | 61 ± 12 | 61 ± 12 | 86 | 83 | 80 | 0.998 |
| diaminopyrimidine | trimethoprim | 51 ± 12 | 49 ± 12 | 59 ± 16 | 77 | 81 | 75 | 0.998 |
| fluoroquinolone | ciprofloxacin | 90 ± 8 | 91 ± 12 | 99 ± 11 | 99 | 112 | 125 | 0.998 |
| | enrofloxacin | 77 ± 11 | 80 ± 15 | 81 ± 11 | 86 | 80 | 75 | 0.998 |
| | moxifloxacin | 83 ± 15 | 81 ± 11 | 87 ± 7 | 90 | 85 | 80 | 0.998 |
| | norfloxacin | 69 ± 11 | 77 ± 15 | 81 ± 9 | 83 | 80 | 81 | 0.998 |
| lincosamide | lincomycin | 71 ± 17 | 81 ± 15 | 82 ± 11 | 91 | 87 | 85 | 0.998 |
| macrolides | azithromycin | 69 ± 11 | 71 ± 13 | 68 ± 13 | 86 | 80 | 75 | 0.998 |
| | clarithromycin | 99 ± 11 | 104 ± 9 | 114 ± 9 | 100 | 113 | 122 | 0.998 |
| | clarithromycin N-oxide | 40 ± 16 | 44 ± 11 | 44 ± 14 | 76 | 80 | 75 | 0.999 |
| | erythromycin | 43 ± 11 | 44 ± 12 | 50 ± 9 | 80 | 80 | 76 | 0.998 |

**Table A1.** *Cont.*

| Group | Antibiotic | Recovery (%) | | | Matrix Effect (%) | | | Linearity |
|---|---|---|---|---|---|---|---|---|
| | | 0.01 | 0.05 | 0.1 | River Water | Outlet-WWTP * | Inlet-WWTP * | |
| sulfonamide | dapsone | 70 ± 18 | 75 ± 14 | 72 ± 17 | 90 | 100 | 107 | 0.998 |
| | sulfacetamide | 91 ± 7 | 92 ± 11 | 91 ± 8 | 89 | 90 | 78 | 0.998 |
| | sulfadiazine | 92 ± 7 | 93 ± 11 | 94 ± 10 | 87 | 88 | 80 | 0.998 |
| | sulfadoxine | 93 ± 12 | 93 ± 10 | 90 ± 10 | 87 | 80 | 78 | 0.998 |
| | sulfamerazine | 90 ± 7 | 89 ± 10 | 89 ± 7 | 91 | 90 | 87 | 0.998 |
| | sulfapyridine | 95 ± 5 | 90 ± 11 | 93 ± 13 | 92 | 89 | 77 | 0.998 |
| | sulfamethoxazole | 91 ± 10 | 95 ± 5 | 95 ± 10 | 97 | 90 | 91 | 0.998 |
| | sulfamethoxypyridazine | 99 ± 7 | 95 ± 11 | 94 ± 13 | 95 | 90 | 89 | 0.998 |
| tetracycline | oxytetracycline | 91 ± 11 | 90 ± 10 | 90 ± 10 | 88 | 85 | 80 | 0.998 |

**Table A2.** LOD and LOQ values obtained for the studied antibiotics' methodology.

| Group | Antibiotic | LOD μg L$^{-1}$ | LOQ μg L$^{-1}$ |
|---|---|---|---|
| florfenicol metabolite | florfenicol amine | 0.093 | 0.306 |
| ß-Lactamase | amoxicillin | 0.152 | 0.500 |
| | ampicillin | 0.021 | 0.071 |
| ß-Lactamase AMX degradation product | diketopiperazine | 0.202 | 0.610 |
| | penicilloic acid | 0.151 | 0.460 |
| diaminopyrimidine | trimethoprim | 0.026 | 0.085 |
| fluoroquinolone | ciprofloxacin | 0.285 | 0.094 |
| | enrofloxacin | 0.041 | 0.135 |
| | moxifloxacin | 0.011 | 0.035 |
| | norfloxacin | 0.264 | 0.871 |
| lincosamide | lincomycin | 0.025 | 0.082 |
| macrolides | azithromycin | 0.020 | 0.067 |
| | clarithromycin | 0.018 | 0.059 |
| | clarithromycin N-oxide | 0.012 | 0.038 |
| | erythromycin | 0.024 | 0.072 |
| sulfonamide | dapsone | 0.029 | 0.095 |
| | sulfacetamide | 0.049 | 0.163 |
| | sulfadiazine | 0.046 | 0.152 |
| | sulfadoxine | 0.012 | 0.038 |
| | sulfamerazine | 0.230 | 0.750 |
| | sulfapyridine | 0.029 | 0.097 |
| | sulfamethoxazole | 0.008 | 0.025 |
| | sulfamethoxypyridazine | 0.060 | 0.197 |
| tetracycline | oxytetracycline | 0.028 | 0.094 |

Abbreviations: LOD = limit of detection, LOQ = limit of quantification.

**Table A3.** ANOVA test between Spanish and French rivers.

| Sources of Variation | Sum of Squares | Degree of Freedom | Mean Square | Factor F | *p*-Value |
|---|---|---|---|---|---|
| Between rivers | 21.642.135 | 1 | 21.642.135 | F = 5.27929 | 0.024 |
| Within rivers | 344.353.214 | 84 | 4.099.443 | | |
| Total | 365.995.349 | 85 | | | |

**Table A4.** Antibiotic maximum concentrations in Spain and France.

| Group | Antibiotic | $C_{max}$ Spain (ng L$^{-1}$) | Reference | $C_{max}$ France (ng L$^{-1}$) | Reference |
|---|---|---|---|---|---|
| ß-Lactamase | amoxicillin | n/d | | 68 | [71] |
| | ampicillin | n/d | | n/d | |
| fluoroquinolone | ciprofloxacin | 740 | [44] | 9660 | [42] |
| | enrofloxacin | 178 | [31] | n/d | |
| | moxifloxacin | 205 | [72] | n/d | |
| | norfloxacin | n/d | n/d | 163 | [45] |
| lincosamide | lincomycin | 47 | [44] | n/d | |
| macrolides | azithromycin | 28 | [71] | n/d | |
| | erythromycin | 70 | [72] | 4 | [71] |
| | clarithromycin | 91 | [44] | 2330 | [42] |
| sulfonamide | sulfadiazine | 2312 | [73] | n/d | |
| | sulfamerazine | n/d | | n/d | |
| | sulfamethoxazole | 11,000 | [73] | 544 | [45] |
| | sulfapyridine | 12,000 | [74] | 1 | [75] |
| diaminopyrimidine | trimethoprim | 252 | [76] | 20 | [75] |

Abbreviations: $C_{max}$ = maximum concentration.

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
