# Peer review of "Screening for Antibiotics and Their Degradation Products in Surface and Wastewaters of the POCTEFA Territory by Solid-Phase Extraction-UPLC-Electrospray MS/MS"

_water, doi:10.3390/w15010014_

Round 1

Reviewer 1 Report (Previous Reviewer 1)

The authors have significantly improved the analytical quality information of the Manuscript. However, there are still some comments to be addressed.

- In analytical chemistry, all m/z values when talking about high resolution mass spectrometry must be expressed with 4 decimals. 

- Previous comment 3.2: the authors have included reference 37, and they state that the tentative identification level of the compounds found after the developed screening strategy is 2. As the authors may know, the identification levels 2a and 2b needs data of the MS2 spectra to be checked. Which parameters were set for library identification to accomodate the tentatively identified compounds in this level? Please, include this information in section 2.5.

- Previous comment 3.3 and Table S1: recoveries must include RSD values expressed as %. Matrix effects shown in Table S1 are extremely high, showing in all cases strong matrix effects. Can the authors define how matrix effect tests were performed and how were these values calculated?

- Previous comment 5.1 and Table 5: please, include: adduct observed, exact m/z value expressed with 4 decimals, and the identification confidence level of each compound (specify 2a or 2b). 

- Please, check in the whole Manuscript the use of the word "metabolised" if it is not referred to a specific metabolic pathway of a (micro)organism. Suggestion: transformed.

- Table 6: under the opinion of this reviewer, transformation products of amoxicillin should not be included since no information is given in this table.

- Previous comment 13: precisely. Due to it was not possible to acquired their analytical standards, for a reliable tentative identification criteria, an identification confidence level must be use. Please, follow the recommendations of reference 37 and other literature focused on the identification of unknowns to categorise each transformation product.

Author Response

Find attached your answers.

Reviewer 2 Report (New Reviewer)

English language and style are fine but minor spell check required, for example:

- line 122: degree sign underlined.

- 178: CE was doubled.

- 204: other authors.

-  309, 337, 338: ca.??????

-415: -lactamases??

------------

-Abbreviations should be identified especially in tables.

- correct the Y axis title in Fig 4.

- the title of Y axis is missing in both Fig 2 and fig 3.

- Authors should include some of future perspectives and recommendation based on their results. they mentioned that there is a contamination with antibiotics with different levels, but what is the solution they are providing for that situation?

- Is it possible to remove the abbreviations from the title?

Author Response

Find attached the answers to your comments.

Round 2

Reviewer 1 Report (Previous Reviewer 1)

The authors have corrected major reviewer's comments.

This manuscript is a resubmission of an earlier submission. The following is a list of the peer review reports and author responses from that submission.

Round 1

Reviewer 1 Report

This manuscript deals with the detection and quantification of a variety of antibiotics in surface waters and WWTP effluents by LC-HRMS. Although the area of study could be of interest, the manuscript lacks of novelty and soundness in all terms because the intended topic has been studied from the last 20 years worldwide. Besides, the analytical approaches are scarcely shown and this points out the lack of analytical quality of the results. The authors address in many cases wrong statements and analytical assumptions. Furthermore, a study of the toxicological implications of the presence and persistance of antibiotics in aqueous environments is not disscussed, which would increase the impact of this study.

Please, find a list of recommendations:

1. Reference format is not adequate. Several examples can be found in the whole text. References 24 and 25 are repeated. 

2. Introduction: Not a single mention to the EU Watch List of substances for its monitoring in surface water for EU members, which would be the basis and the main reason of the study... Please, note that there is a new release of this list published in July 2022. In this list, there are some antimicrobials such as clindamycin and ofloxacin for which data in this study is not included. Although the authors lack of their analytical standards, were not these two compounds detected in the screening strategy? Please, check.

3. The authors state in objective i) of this study "to develop a method for the simultaneous analysis of a large spectrum of antibiotics together with their metabolites at the detection limits allowing the screening of surface waters". Which is the novelty of your extraction and analysis methodology in comparison with those publised from the last 20 years and which are mentioned in the text and referenced?

Method validation merits a paragraph in materials and methods section, as well as a proper validation results table must be included (even in supplementary information) with the following information: individual recoveries for each compound at a low and a medium concentration levels with the corresponding relative standard deviations (omitted in the whole text), matrix effect, repeatability,  reproducibility, linearity, LOQs and LODs. The authors show LODs and LOQs for the target compounds but they do not mention if these limits fullfill the identification requirements for Q-Orbitrap analysers reported in SANTE Guidelines or any other Guidelines.

No analytical quality guidelines have been followed to ensure the analytical quality of the results presented in this study.

The authors state in lines 256-257 that recoveries were performed at three concentration levels, but later on, in lines 257-259 they mention a single concentration level at 500 ng/L. Which are the concentrations that recoveries were carried out through the whole SPE and LC-MS/MS method? Were they replicated? Please, clarify.

This reviewer is quite intringued about the recovery results of amoxicillin with the applied SPE method using HLB cartridges without sample pH adjustment.

Have the authors taken into account that amoxicillin is degraded in methanol stock standard solution by means of a week? The quantification results of amoxicillin are likely to be those of its main degradation product, which m/z values and product ions are the same as amoxicillin but normally elutes at a different retention time. Please, clarify which compound was quantified.

4. It is not clear if the authors developed a PRM method for target quantification and then a DDA was performed for the suspect screening. If so, a DDA with an include list of the target compounds could be developed and all the information would be available in a single run. Please, clarify.

Which software was used for target quantification purpuses? It is not mentioned in the text. Please, describe.

5. The authors performed as well a suspect screening strategy using a  database of the main degradation products reported in literature. However, the authors do not give details of the analytical confidence identification levels of the suspects found, although they state that 3 of them were confirmed with the retention time of the analytical standard. Information about match of the isotopic profile, number of product ions and mass error considered for tentative candidates, comparison of MS/MS spectra with libraries or literature is not included in materials and methods section neither discussed in results section. Please, revise.

Were all the transformation products included Table 5 confirmed with the retention time of the analytical standard? Please, check the following reference for a proper discussion of the suspect screening results: E. Schymanski et al. Environ. Sci. Technol. 2014, 48, 4, 2097-2098.

6. Figure 2: elution of compound 1 is not gaussian. Was the chromatography optimised to solve this elution behaviour? Please, clarify.

7. Figure 3 a) and b): why does the chromatographic peak that elutes at 10 min seem "cutted" in the baseline? Please, explain.

8. ANOVA test is not described in Materials and Method section. This reviewer could not access to supplementary information to check the ANOVA results.

9. Tables 2 and 3 can be merged. Maybe, information of Table 2 would merit a Figure to help the reading of the results.

10. Results and discussion sections are devoted to the quantification and frequency of detection of the target/suspect compounds, and the comparison with other studies. However, the results do not highlight the importance of detecting antibiotics and their by-products in environmental waters neither specifies the (eco)toxicological and or environmental antibiotic resistantant spread risks. It is opinion of this reviewer that a discussion in this line would be interesting and would support the impact and novelty of the study rather than a plain comparison with other studies conducted 15 years ago, the scientific community already knows the presence of antibiotics in environmental compartments.

11. Lines 413-415: why is it important to determine and monitor antibiotic degradation products that do not inherent antibiotic activity? Please, clarify.

12. Lines 416-417: what was the methodology followed for the quantification of these two transformation products? Was the developed methodology especifically validated for them after compound confirmation? If so, please, include their validation results in the validation table requested above. Otherwise, clarify how they were quantified.

13. Table 5. Please, include the identification confidence level for each compound using the provided reference of comment 5.

14. Conclusions: please, revise the text and include the main conclusions of antibiotic resistance spread results.

Reviewer 2 Report

Dear Editor,

Thank you for the opportunity to review the manuscript entitled Screening for antibiotics and their degradation products in surface and wastewaters of the POCTEFA territory by solid-phase extraction – UPLC- electrospray MS/MS. This manuscript is written in details in scientific manner. The presented research is of high importance and the conducted method is novel and applicable in other water samples. The methodology is well-composed and all experiments are conducted in a right manner. Also, the used references are appropriate. Just a few things should be corrected:

In line 23 there is an extra e in the

After the Table 1 an explanation of aberrations presented in the Table should be added.